# Development and Validation of a Lifestyle Behavior Tool in Overweight and Obese Women through Qualitative and Quantitative Approaches

**DOI:** 10.3390/nu13124553

**Published:** 2021-12-20

**Authors:** Chee Wai Ku, Rachael Si Xuan Loo, Cheryl Jia En Lim, Jacinth J. X. Tan, Joey Ee Wen Ho, Wee Meng Han, Xiang Wen Ng, Jerry Kok Yen Chan, Fabian Yap, See Ling Loy

**Affiliations:** 1Duke-NUS Medical School, 8 College Road, Singapore 169857, Singapore; jerrychan@duke-nus.edu.sg (J.K.Y.C.); fabian.yap.k.p@singhealth.com.sg (F.Y.); 2Department of Reproductive Medicine, KK Women’s and Children’s Hospital, Singapore 229899, Singapore; ng.xiang.wen@kkh.com.sg; 3Department of Paediatrics, KK Women’s and Children’s Hospital, Singapore 229899, Singapore; rachael.loo.s.x@kkh.com.sg; 4Yong Loo Lin School of Medicine, National University of Singapore, Singapore 119228, Singapore; e0199411@u.nus.edu; 5School of Social Sciences, Singapore Management University, Singapore 178903, Singapore; jacinthtan@smu.edu.sg; 6Department of Dietetics, KK Women’s and Children’s Hospital, Singapore 229899, Singapore; ho.ee.wen@kkh.com.sg (J.E.W.H.); han.wee.meng@kkh.com.sg (W.M.H.); 7Lee Kong Chian School of Medicine, Nanyang Technological University, Singapore 636921, Singapore

**Keywords:** obesity, lifestyle behavior tool, 6P, healthy nutrition

## Abstract

There is a paucity of effective intervention tools for overweight/obese women to assess, guide and monitor their eating behavior. This study aimed to develop a lifestyle intervention tool, assess its acceptability and usefulness, and verify its construct validity in overweight/obese women. The 6P tool (Portion, Proportion, Pleasure, Phase, Physicality, Psychology) was developed and 15 women with a body mass index (BMI) ≥ 25 kg/m^2^ were interviewed to assess its perceived acceptability and usefulness. Subsequently, the revised 6P tool was tested in 46 women with a BMI ≥ 25 kg/m^2^. The Three-Factor Eating Questionnaire (TFEQ), International Physical Activity Questionnaire-Short (IPAQ), and weight were measured at baseline and one-month. Most participants were satisfied with the presentation of the 6P tool (86.8%), and agreed it was useful in guiding healthy eating (81.6%) and raising awareness of eating behavior (97.4%). There were significant improvements in cognitive restraint (*p* = 0.010) and disinhibition (*p* = 0.030) (TFEQ), portion size (P1), pleasure behaviors (P3), and total composite 6P score (*p* < 0.001). However, there was no significant reduction in weight or increase in physical activity. The 6P tool is acceptable and presents with good validity for assessing lifestyle behaviors.

## 1. Introduction

The worldwide prevalence of obesity in women of reproductive age is increasing [1]. Not only does obesity increase the risk of chronic diseases [2,3], it also increases the risk of infertility, pregnancy complications, and the long-term risk of cardiometabolic and neurodevelopment disorders in their children [1,4,5]. With heightened awareness of the intergenerational impact of maternal obesity, this issue has come to the forefront of scientific research and led to the development of strategies to address this major health challenge. Although high quality evidence of the effectiveness of preconception interventions on pregnancy outcomes is limited [6], adopting a life-course approach with behavioral interventions, particularly the promotion of healthy eating throughout the preconception, pregnancy, and postpartum period, could potentially improve maternal-child wellbeing and break the vicious cycle of obesity, thus preventing its complications [7,8]. 

The fundamental cause of obesity is an energy imbalance between calories consumed and calories expended, contributed to by a combination of unhealthy eating habits, physical inactivity, and reduced energy expenditure [3]. Interventions that combine eating and exercise behaviors are often used to alleviate weight problems [3,9,10]. However, these weight loss interventions have generally failed to induce a sustainable behavior change and, even if effective, they generally resulted in only small changes in target behaviors [11,12]. A systematic review reported the evidence base for using behavior change theories or techniques to inform the development of diet and nutrition interventions, to be fair in our evaluation [13]. The relatively short duration of the intervention may be the reasons for its inability to produce any meaningful impact. Furthermore, practical barriers such as cost and intensive coaching over a long period of time may pose as a challenge to effective behavioral intervention. Creative routes are therefore needed to engage individuals to undertake sustained lifestyle changes that are both acceptable and cost-effective. Evidence suggests that identifying and prioritizing specific factors that influence healthy eating would offer more prospects for successful weight loss and metabolic control than a weight-loss-centred approach [14,15,16,17,18]. It is therefore important to develop a behavioral intervention tool that can be independently used by individuals to promote their healthy eating habits and physical activity, and to sustain motivational level in the long term. In so doing, this comprehensive lifestyle intervention will seek to improve metabolic health and fertility, reduce pregnancy complications, and improve cardiometabolic and neurodevelopmental outcomes in the children of obese women. 

In this study, our aim is to introduce a newly developed tool, named the 6P tool, which is designed based on the mental model principle to promote healthy eating and increase physical activity, to achieve caloric balance. The mental model is a conceptual framework or cognitive structure held in an individual’s mind that shapes the way the person perceives the world based on personal experience and understanding, and how they act upon them [19,20]. The mental model is used to reason and make decisions, and can be the basis of individual behavior as it exists in memory [21]. In the specific case of mental models for nutrition, the underlying knowledge structure of how diet and nutrition are related to obesity may be useful for encouraging and establishing sustainable healthy eating habits. Furthermore, rationalizing and breaking down the components that make up healthy eating patterns or behaviors also allow an individual to take control of and improve their current eating habits. The 6P tool conceptualizes nutrition and activity as six discrete components: Portion, Proportion, Pleasure, Phase, Physicality, and Psychology. The 6P framework provides a knowledge structure to facilitate an individual’s understanding of healthy nutrition and the relationships between the ‘Ps’, to self-evaluate deviant and unhealthy practices, and to make decisions about a specific component of the ‘Ps’ for action. The 6P tool focuses on promoting self-awareness of unhealthy lifestyle behaviors, nudging individuals into actual implementation via concrete, personalized feedback, and increasing intrinsic motivation. This is based on the theoretical framework of the Theory of Planned Behavior (TPB) [22], which has been widely used to predict behavioral intention and health-related behaviors [22,23]. There are three main constructs in the TPB that contribute to behavioral intention, which is the strongest predictor of behavior [24,25]. Firstly, attitude, which comprises knowledge, beliefs, and values towards a behavior, in either a positive or a negative light. Secondly, subjective norms describe the perceived social pressure to perform a desirable behavior. Thirdly, perceived behavioral control refers to the amount of control an individual feels she has over performing a behavior [22]. 

Therefore, we qualitatively evaluated the acceptability and perceived usefulness of the 6P tool in overweight and obese women at different stages of life, that is, before, during, and after pregnancy. Subsequently, we validated the 6P tool by evaluating pre-post intervention changes in 6P scores, eating behavior, physical activity, and body weight, among an independent group of overweight and obese women. This 6P tool is intended to be subsequently applied as part of a behavioral intervention in a cohort of overweight and obese women throughout their preconception, pregnancy, and postpartum phases.

## 2. Materials and Methods

### 2.1. Study Design

Standardized methodology was applied in the process of developing and validating the 6P tool, including a literature review, group discussion, designing the content, rough draft development, expert evaluation, pre-testing, and a pilot study for face, content, and construct validity assessment [26,27]. The workflow of the 6P development and validation procedures are shown in Figure 1, starting from February 2020 until September 2021. The study was carried out at the KK Women’s and Children’s Hospital (KKH), Singapore. This study was conducted according to the guidelines laid down in the Declaration of Helsinki. Ethics approval was obtained from the SingHealth Centralized Institute Review Board (reference 2020/2530), Singapore. All participants provided their informed consent in writing.

### 2.2. Defining Scope, Structure, and Content

A comprehensive review of the literature and multiple discussions among relevant domain experts, comprising obstetricians, endocrinologists, and dietitians, were carried out to identify the concept, items, and content necessary for inclusion in the 6P tool. The tool was formulated with a holistic approach of dietary management by covering energy intake (P1 Portion, P2 Proportion, P3 Pleasure, P4 Phase), energy expenditure (P5 Physicality), and motivation (P6 Psychology) as the basis to make dietary changes. Figure 2 shows the description of each component in the 6P. These components were determined from common dietary and lifestyle issues related to weight gain and BMI in the local population, including excessive carbohydrate intake, lack of dietary fiber intake, regular fast-food intake, unhealthy snack and sweetened beverage intake, predominantly night-time eating, and sedentary behavior [28,29,30,31,32,33]. 

#### Expert Evaluation

The initial draft was sent to a group of multidisciplinary experts involved in the care of overweight and obese women for critical appraisal. Five independent clinicians and researchers from the field of Obstetrics and Gynaecology, Dietetics and Nutrition, and Women’s Health tested the 6P tool for face and content validity, resulting in further refinement. To sustain behavioral changes, dietitians developed and reviewed short health messages containing 6P recommendations, which serve as nudges to remind and motivate participants during the follow-up period. 

### 2.3. Pre-Testing (Study 1)

The 6P tool was pre-tested among 15 women with BMI ≥ 25 kg/m^2^ between September 2020 and December 2020. These women were at different stages of life, either in the preconception, pregnancy, or postpartum phase. Researchers conducted a face-to-face audio-recorded interview to obtain participants’ comments on the 6P tool. The aim of the interview was to verify face validity, improve the presentation and language of the tool, and to determine the understanding and usefulness of the tool for participants in the preconception, pregnancy, and postpartum phases. Samples of short text messages on 6P recommendations were shown to the participants to obtain feedback on the messages’ content, presentation, and usefulness, as well as the preferred frequency and delivery mode of these messages. Data were analysed qualitatively and the 6P tool and messages were revised accordingly. Based on the feedback from the participants, we developed a list of health implications and recommendations based on the dietary problems derived from the 6P, which formed the basis for a digital platform for delivery. The recommendations were based on the World Health Organization’s general nutrition guidelines [34], the Singapore Dietary Guidelines [35,36], and research evidence [37,38,39,40,41]. A recent meta-analysis has suggested that the optimal number of recommendations for behavior change is between two and three [42], and, hence, the number of recommendations for each dietary problem was limited to three.

### 2.4. Pilot Study (Study 2)

The revised version of the 6P tool, incorporating health implications and recommendations on a digital platform, was tested among 46 women with BMI ≥ 25 kg/m^2^ who were planning to conceive, between July 2021 to September 2021 (Appendix A). The study was registered on ClinicalTrials.gov (NCT04582643). At the time of recruitment, demographic and general lifestyle information were collected, and weight and height were recorded under the direction of a research staff or on-duty nurse in the clinic using a digital weighing machine (Avamech B1000-M, Singapore). BMI was calculated as weight in kilograms divided by height squared in meters (kg/m^2^). Eating behavior was assessed using the Three-Factor Eating Questionnaire (TFEQ-R51), which covers three domains of eating behavior: ‘cognitive restraint’ (21 items), ‘disinhibition’ (16 items), and ‘hunger’ (14 items) [43]. Each question is scored 0 or 1 and summed. Higher scores in the respective scales indicate greater cognitive restraint, disinhibition, and predisposition to hunger, respectively [43]. The TFEQ had good internal consistency reliability coefficients for the three subscales [44] and has been widely used in clinical setting to study the eating behavior of obese individuals [45,46]. The Cronbach’s alpha in this study was 0.78 for the cognitive restraint domain, 0.68 for the disinhibition domain, and 0.70 for the hunger domain. Physical activity was assessed using the International Physical Activity Questionnaire-Short (IPAQ) [47]. Data were calculated as metabolic equivalents (MET-minutes/week) scores and categorized as “inactive” (≤599 MET-minutes/week), “minimally active” (600–2999 MET-minutes/week), or “highly active” (≥3000 MET-minutes/week) [47].

The 6P tool was administered online under the guidance of the research staff. Participants were asked about their dietary behavior in the past week, on weekdays (working days), and on weekends (non-working days). A sample of the 6P tool is displayed in Appendix A. Transformation of the 6P tool into the digital platform allowed an individualised 6P feedback report and monitoring chart to be generated in real-time. The feedback report contained information on the health implications and recommendations based on their specific dietary problems (Appendix A), while the monitoring chart facilitated participants to track their 6P results over time. From the list of identified dietary problems, participants were prompted to select two items from the 6P as their goals to make changes according to their own capability. The workflow from administering the 6P tool to receiving 6P mobile messages by participants is shown in Figure 3.

Participants were followed up for one month, and image-based health messages on 6P recommendations were sent to their mobile phone twice every week (Wednesday and Saturday). These personalized mobile health messages were in line with their goal selection. Samples of the 6P mobile health messages are illustrated in Appendix A. During the 1-month follow-up visit, all participants were re-assessed for their weight, physical activity, eating behavior, and 6P responses. They were also asked to complete an evaluation questionnaire on a 5-point Likert scale to assess the clarity of each 6P item, usefulness of the 6P tool, and respective mobile health messages.

### 2.5. Statistical Analysis

For the 6P pre-testing (Study 1), transcripts were imported into NVivo, Release 1.4 (QSR International) to facilitate electronic coding and data retrieval. Data coding and analysis followed both deductive and inductive approaches [48,49]. Two researchers independently reviewed the transcripts and developed an initial set of themes from the interview topics, with additional codes that emerged from the data. After developing the initial coding scheme, the two researchers independently performed the coding analyses. Minor discrepancies were detected and resolved through discussion. A third researcher reviewed the transcripts to ensure that the themes were reflective of the data.

For the pilot study (Study 2), continuous variables were presented as means and standard deviations (SDs), or as medians and interquartile ranges; while categorical variables were presented as numbers and percentages. The internal consistency of the 6P questions was determined using the Cronbach’s alpha. A composite score was determined for each P and a total composite score was derived by summation of the scores for each P (Appendix A). The associations of 6P scores with eating behavior, physical activity, and weight were examined using the Pearson correlation or Spearman’s rank correlation, as appropriate. The pre-post changes in the 6P scores, eating behavior, physical activity, and weight were analyzed using the paired *t*-test or Wilcoxon sign rank test for continuous variables and McNemar’s test for categorical variables. Statistical analyses were conducted using the SPSS Statistics Version 20 (IBM Corp, Armonk, NY, USA) and Stata Statistical Software, Release 16 (StataCorp, College Station, TX, USA).

## 3. Results

### 3.1. Pre-Testing (Study 1)

#### 3.1.1. Characteristics of the Participants

Of the 15 women who participated in the interview, 60% were Chinese (9/15), 13% were Malay (2/15), and 27% were Indian (4/15). Most had attained tertiary education (10/15, 67%), were employed (11/15, 73%), were non-smokers (11/15, 73%), and reported never or rarely having consumed alcohol (12/15, 80%). The mean age was 33.8 years (SD 5.1), while the mean BMI was 33.7 kg/m^2^ (SD 5.7).

A total of six themes emerged from the evaluation process of the participants’ views on the 6P tool and the 6P health messages. Examples of quotes from participants for each theme are documented in Table 1. 

#### 3.1.2. Perceived Acceptability of the 6P Tool

Overall, the content and language of the 6P tool were commented on positively by the participants, in terms of comprehensiveness and clarity. Meanwhile, suggestions were provided to improve the estimation of the portion size. In terms of delivery platform, the majority of the participants preferred to fill up the 6P tool on a mobile health platform instead of using the hardcopy. The participants also showed willingness to administer the 6P tool weekly or even daily.

#### 3.1.3. Perceived Relevance and Usefulness of the 6P Tool

The participants shared that the questions asked in the 6P tool were very relevant to their eating habits. They indicated that administrating the 6P tool could increase their knowledge about healthy eating, help them to be aware of unhealthy eating habits, and facilitate and monitor their behavior changes.

#### 3.1.4. Adjuncts to the 6P Tool

The participants also commented that it would be helpful to provide a feedback report relating to their 6P performance and recommendations, giving them a specific direction to address their dietary problems. Some participants would like to have a 6P tracking chart which can aid in behavior change.

#### 3.1.5. Perceived Acceptability of Mobile Health Messages

The participants were open to and expressed interest in the idea of text messaging to promote healthy eating. They were satisfied with the content and language used. A mobile application was frequently mentioned as a preferable delivery platform for messages to be sent weekly.

#### 3.1.6. Perceived Usefulness of Mobile Health Messages

The participants shared that the messages provided new knowledge about healthy eating and could remind them of their dietary goals. Some said they knew much of the information contained in the text messages but felt that it was a useful reminder and motivated renewed attention to eating choices.

#### 3.1.7. Suggestions for Improving Mobile Health Messages

The participants also shared that too many words in the messages could lead to boredom and the messages ignored by users. In addition to traditional SMS text messages, participants suggested including images which are designed based on local culture. They were interested in receiving texts related to foods to be taken and avoided.

### 3.2. Pilot Study (Study 2)

#### 3.2.1. Characteristics of Participants

In total, 73 women were screened for eligibility. Of these, 46 women with a BMI ≥ 25 kg/m^2^, between 21 and 45 years of age, and planning to conceive were recruited during the preconception period, while 27 women were not eligible or declined to participate (Appendix A). The mean age was 33.70 (SD 4.77) years. The participants were mostly Chinese (25/46, 54.4%), 26.1% were Malay (12/46), and 15.2% were Indian (7/46). Most were employed (42/46, 91.3%), non-smokers (43/46, 93.5%), and reported never or rarely having consumed alcohol (42/46, 93.3%). About half of them attained tertiary education (22/46, 47.8%) and did not have underlying medical conditions (24/46, 52.2%). Nine participants (19.6%) had an underlying metabolic disorder (diabetes mellitus / hypertension / hyperlipidemia). Most of the participants did not smoke (43/46, 93.5%) or drink alcohol (42/46, 93.3%). 

#### 3.2.2. Internal Consistency of 6P

The 6P tool consisted of two sections: first, the main questionnaire to quantitatively assess eating patterns, activities, and motivational levels related to each of the 6Ps, and, secondly, a set of accompanying self-assessment questions. Two self-assessment questions were removed due to poor internal consistency, namely ‘I usually finished everything on my plate, or the portion given to me’ and ‘I was aware of the barriers to change.’ The Cronbach’s alpha for the final 6P tool consisting of 31 items was 0.75, which indicated an acceptable level of the 6P tool’s reliability [50]. 

#### 3.2.3. Associations of 6P Scores with Eating Behavior, Physical Activity, and BMI

The total composite score of 6P was positively correlated with the two subscales of TFEQ, namely disinhibition (r = 0.50, *p* < 0.001) and hunger (r = 0.45, *p* < 0.002), and these were negatively correlated with cognitive restraint (r = −0.33, *p* = 0.027) (Table 2). There was a negative correlation between P5 Physicality composite score and IPAQ total MET (r = −0.52, *p* = 0.002). A positive correlation trend was observed between the 6P total composite score and BMI (r = 0.26, *p* = 0.086). 

#### 3.2.4. Weight, Eating Behavior, and Physical Activity before and after the Intervention

A total of 38 participants completed follow-up at one month. As shown in Table 3, there was no significant weight change (76.28 kg, SD 11.29 vs. 76.37 kg, SD 11.20, *p* = 0.719) or change in BMI (30.32 kg/m^2^, SD 4.09 vs. 30.36 kg/m^2^, SD 4.10, *p* = 0.682) before and after the 6P intervention for a period of one month. There was a significant improvement in two dietary behavior domains as measured by TFEQ, namely cognitive restraint (10.26, SD 4.10 vs. 11.82, SD 3.83, *p* = 0.010) and disinhibition (7.63, SD 2.87 vs. 6.68, SD 2.90, *p* = 0.030) but not hunger (4.92, SD 2.88 vs. 4.50, SD 2.98, *p* = 0.293). However, no changes in physical activity level were detected before and after the intervention.

#### 3.2.5. 6P Assessment before and after Intervention 

There was a significant decrease in portion size (P1), from the median portion size of 3.00 (2.00–4.50) to 2.63 (1.25–3.67) (*p* = 0.004). The recommended portion size is between 1–2, with 13 (34.2%) participants achieving this target post-intervention, compared to 6 (15.8%) pre-intervention (*p* = 0.039). There was a trend towards improvement in P4 Phase (from 25% to 20%) and P5 Physicality (from 195 min per week to 285 min per week) at the one-month follow-up. The vegetable intake (P2), frequency of pleasure food intake (P3), and motivational level (P6) were similar before and after the intervention (Table 3).

#### 3.2.6. 6P Composite Scores before and after Intervention

A composite score was determined for each P to facilitate comparison before and after the intervention. A total composite score was then derived by summation of the scores for each P to provide a simple and convenient summary of the 6P results for both participant and the clinician, and to track the ongoing progress in their nutritional behaviors. The composite score for P1 Portion was reduced from a mean score of 3.55 (SD 1.08) to 2.87 (SD 1.33) (*p* = 0.002) at the one-month follow-up visit. Similarly, the composite score for P3 Pleasure was reduced from a mean score of 5.89 (SD 2.00) to 4.76 (SD 1.91) (*p* = 0.004) over the one-month period. All 6Ps were summated to give a total composite score of up to 43, where the higher the scores, the greater the tendency to practice unhealthy eating behaviors. A reduction in total composite 6P scores was observed after the one-month intervention, from 18.42 (SD 4.81) to 15.61 (SD 4.98) (*p* < 0.001) (Figure 4).

#### 3.2.7. 6P Diagnosis and Goal Selection

The three main diagnoses of the 6P tool at baseline were P3 Pleasure and Meal Regularity (37/37, 100%), P1 Portion (36/37, 97.3%), and P2 Proportion (35/37, 97.3%). The goals selected most frequently at baseline were P1 (17/37, 45.9%) and P3 (14/37, 37.8%). The three main diagnoses of the 6P tool at follow-up were P3 Pleasure and Meal Regularity (35/37, 92.1%), P2 Proportion (35/37, 92.1%), and P1 Portion (34/38, 89.5%) (Table 4).

#### 3.2.8. 6P Feedback and Evaluation

The feedback survey on the 6P tool showed that it was well received among our study participants (Appendix A). Most of the participants were satisfied with the overall presentation of the 6P tool (33/38, 86.8%). They agreed that the 6P tool was useful in guiding healthy eating (31/38, 81.6%), raising awareness of eating behavior (37/38, 97.4%), and providing new knowledge related to healthy eating (33/38, 86.8%). Five (13.2%) participants suggested improvements to the 6P tool, including improvements to the interface, clearer definitions in the questions, and clearer explanations in the feedback report. The mobile health messages were well received by most of the participants and were considered useful in motivating them to improve their eating habits (33/38, 86.8%), and were sent at a suitable frequency (33/38, 86.8%). Two participants (5.3%) suggested that we include more detailed visuals and explanations. The top main features of the 6P tool that the participants found useful in promoting healthy eating behavior were the content of the mobile health messages matched to their selected goal (25/38, 65.8%), the diagnosis and explanation of nutrition (20/38, 52.6%), and the goal-setting feature (18/38, 47.4%). 

## 4. Discussion

In this study, we developed and validated a novel digital lifestyle behavior intervention tool, known as the 6P tool, to promote healthy eating, exercise, and motivation among overweight and obese women. The 6P tool is designed based on the mental model and TPB model by conceptualizing a healthy nutrition framework into six discrete components, namely Portion, Proportion, Pleasure, Phase, Physicality, and Psychology. Acceptable internal consistency and face, content, and construct validity were obtained. In the first qualitative study, we have shown that the 6P tool and its accompanying mobile health messages were perceived to be acceptable, relevant, and useful. Improvements were made based on the suggestions of the study participants, and a digital version of the 6P tool was used for the second validation study, which incorporated the new feedback, recommendation, and goal setting features, as well as personalized mobile health messages tailored to selected goals. During the one-month period, this resulted in a significant positive change in dietary behaviors assessed by both the TFEQ (cognitive restraint and disinhibition subscales) and the 6P tool (P1 Portion, P3 Pleasure, and total composite scores). 

We found a significant improvement in the TFEQ subscales after the intervention in which the participants displayed a higher level of cognitive restraint and an improvement in disinhibition, as well as an improvement in P1 Portion and P3 Pleasure. This is consistent with other studies which showed that women with greater dietary cognitive restraint and less disinhibition were more likely to report a lower total caloric intake and less frequent consumption of snacks and sugary drinks [51,52]. Improvements in P1 and P3 were consistent with the two main goals participants selected at the beginning of the study (P1: 45.9%) and (P3: 37.8%). Goal setting has been shown to be a key component in promoting dietary and physical activity behavior change in adults [53] and is an important strategy in understanding and researching behavior change [54]. In our study, P1 and P3 might be areas in which participants had a greater perceived behavioral control and thus were most likely to succeed in achieving these goals. Based on the TPB framework, if an individual has a positive attitude toward healthy eating and physical activity, perceives it to be positively viewed by significant others, and feels that she is in control of it, then she will likely intend to adopt healthy eating and physical activity. Thus, the use of the 6P tool promotes self-awareness of unhealthy lifestyle behaviors and allows an individual to set personalized goals with recommendations to promote healthy eating and physical activity behaviors, leading to greater control over their own health and better-informed health-related choices.

In our pilot study, we did not observe a substantial change in weight or physical activity during the one-month period. This could be due to the short period of intervention, and, therefore, it is unlikely to result in any significant weight loss or increase in physical activity. Furthermore, ecological factors play an important role in eating behavior, where limitations due to work and other life commitments can reduce the individual’s cognitive capacity to control food timing and result in irregular meals [55]. The purpose of the 6P tool is to improve eating behaviors and increase physical activity. Weight loss is not advocated as the primary goal; instead, participants are encouraged to focus on their 6P goals, eliminate barriers, and monitor their own behavior to improve overall metabolic health. Intrinsic motivation can help promote sustainable change in health behavior [56]. During the one-month period, we observed an increase in proportion of participants with a motivation level ≥5 from 78.4% to 89.5%. There was a slight improvement in terms of physical activity level, with half of the participants achieving at least 150 min of moderate or 75 min of vigorous exercise per week during the one-month period. Ideally, we recommend repeated 6P assessment to optimize weight and maintain targeted lifestyle behavior in the long term. This ongoing sustained approach of self-monitoring, accompanied by frequent nudges, may be useful in achieving weight loss and other lifestyle goals over time.

Dietary assessment is pivotal in providing personalized dietary advice, and/or to measure the impact of dietary interventions [57,58]. The 6P tool was deliberately designed to be both a dietary assessment tool and an intervention tool. Compared to the existing commonly used dietary assessment tools, such as questionnaires assessing psychological components of dietary behavior or evaluating eating disorders; food frequency questionnaire, dietary recall, or food diary measuring dietary intake and dietary quality [59], the 6P tool offers several advantages. First, the 6P design allows the dietary behavior to be measured qualitatively with straightforward and easier-to-respond-to questions. Current existing instruments mainly represent abstract behavioral situations (e.g., uncontrolled eating, emotional eating) or hard-to-remember dietary related aspects (e.g., excess or lack of specific macro and micronutrients) [60,61]. Second, the 6P tool can be self-administered with minimal nutritional knowledge, making it quick to complete, easy to score, and the interpretation intuitive. Third, the 6P tool evaluates and provides immediate feedback and recommendations on diet behavior in a one-stop-shop manner. A monitoring chart is also included in the 6P tool, allowing participants to keep track of their progress with healthy eating habits over time. Through the application of self-monitoring techniques accompanying frequent nudges from mobile health messages, participants can self-reinforce, modify, and set realistic goals to improve their current eating habits [62,63]. Lastly, the 6P tool was created to provide a framework of healthy nutrition principles, which encourages participants to be aware of their own dietary and physical activity behaviors. This allows them to develop a positive attitude toward recommended behaviors and empower them to make changes with the feedback report, goal setting, and personalized mobile health messages that promote sustainable behavioral change. This workflow leverages on the concept that intention directly predicts behavior, and the translation of intention into behavior, otherwise known as implementation intention, has been shown to be effective in dietary behavior change programs [64], and effectively improve health outcomes [17]. 

Taken together, the 6P tool is a comprehensive healthy lifestyle instrument that not only focuses on energy and nutrient intake (P1 Portion, P2 Proportion, P3 Pleasure), but takes into consideration energy expenditure (P5 Physicality), readiness for behavior change (P6 Psychology), and chrononutrition (P4 Phase). Chrononutrition, defined as the administration of food according to the circadian rhythm of the body, is a new dietary dimension that has been associated with cardiometabolic health and obesity [65]. Current lifestyle interventions for overweight and obese women lack a life-course approach and are resource intensive, relying heavily on intensive coaching interventions. Digital platforms to deliver these lifestyle interventions are therefore ideal to overcome these challenges due to greater accessibility and outreach, especially in developed countries where there is a great magnitude of cell phone penetration. Studies have shown that mobile health applications that aim to advance lifestyle changes and provide maternal healthcare services have been effective [66,67]. In addition, the use of mobile health messages as nudges represents a preferred architecture strategy which is widely used in public policy making, to alter people’s behavior, and to influence decision-making [68]. These nudges, if delivered in a supportive and encouraging manner, could potentially play a significant role in increasing overall motivation and self-esteem, as well as the ability to self-regulate behavioral changes. Thus, the 6P tool, alongside its personalized mobile health messages tailored to an individual’s dietary and physical activity goals, represents a novel approach to revolutionize the delivery of a lifestyle intervention throughout the life-course in a sustainable fashion.

Strengths of this study include the development and validation of a comprehensive lifestyle intervention tool, 6P, based on health behavior theories and conceptualized as a mental model. Theory-driven interventions in dietary behavior are few and far between [69], and a clear understanding of these theories and the underlying factors that constrain behavioral change in the target population allows for an optimal intervention to be developed to address these constraints [70]. Thus, an in-depth qualitative study was conducted to understand the participants’ acceptability of the 6P tool and mobile health messages, before the pilot study was conducted. This allowed modifications based on feedback, including explanation, feedback, and goal setting functions after 6P administration, which participants deemed a positive experience. The post-intervention evaluation survey determined the formative (justifying the need) and process (delivery) aspects of the intervention. Participants responded positively to the use of the 6P tool, as it was easy to use, raised their awareness of eating behaviors, and guided them towards healthy eating. However, we acknowledge that the 6P tool is targeted at the individual level, as opposed to an ecological approach, which accounts for interactions between an individual and their environment. Health is determined by a complex interaction between an individual and her environment that has a great influence on individual behavior [70]. This includes socioeconomic status, wide availability of cheap food sources, lack of an enabling environment, and the impact of health promotion campaigns and government policies [71]. The uptake and acceptance of any intervention, and whether it is culturally appropriate, will have a huge bearing on its sustainability and widespread adoption nationally and internationally [72]. These factors were not evaluated in this current study. Furthermore, the long-term impact and cost-effectiveness of the 6P tool were also not evaluated given the short duration of follow-up. Finally, the pre- and post-test study design may overestimate the benefit of an intervention due to the regression to the mean effect and temporal changes [73]. Further studies, with the addition of a concurrent control group, are needed to evaluate these ecological factors and control for temporal changes and regression to the mean effects. 

## 5. Conclusions

We present a novel interactive lifestyle intervention tool called 6P, which is based on the mental model and TPB framework to raise awareness of eating habits and activity, and to promote self-directed behavioral change. This involves real-time feedback, goal setting, and personalized mobile health messages. The 6P tool is an easily accessible, feasible, and acceptable intervention with the potential to be both scalable and cost-effective. One-month follow-up with personalized mobile health messages aligned with respective 6P goal setting has shown an overall positive change in dietary habits. Future studies are required to monitor and evaluate the long-term effectiveness of the 6P tool in larger populations.

## Figures and Tables

**Figure 1 nutrients-13-04553-f001:**
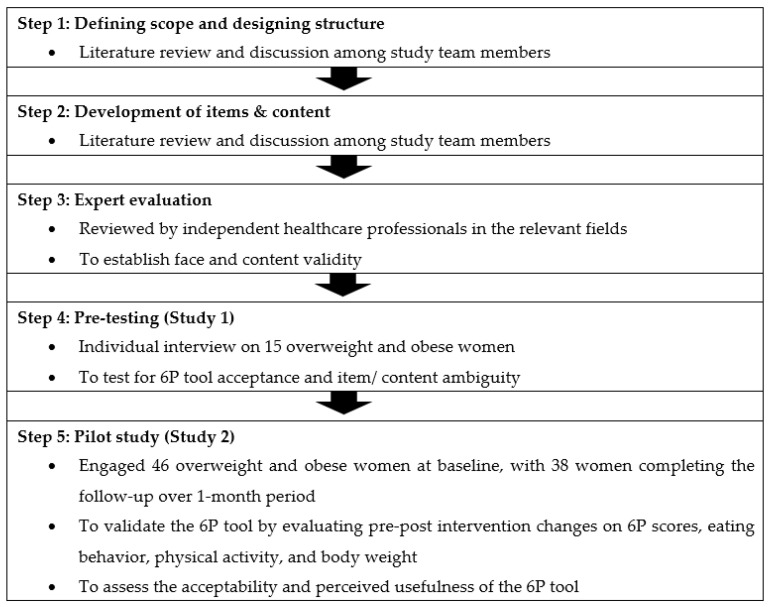
Steps involved in the development and validation of the 6P tool.

**Figure 2 nutrients-13-04553-f002:**
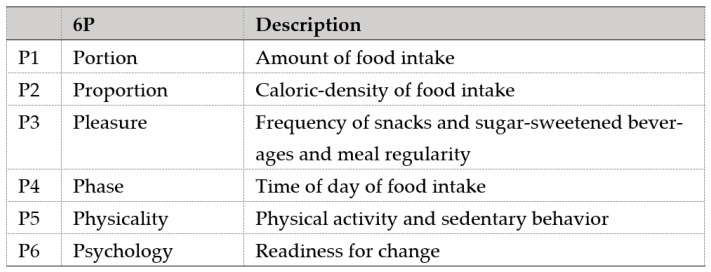
Components of the 6P tool.

**Figure 3 nutrients-13-04553-f003:**
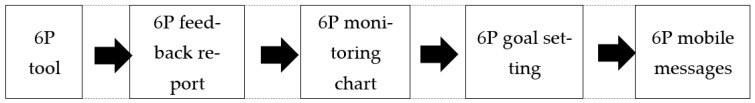
Components of the 6P tool.

**Figure 4 nutrients-13-04553-f004:**
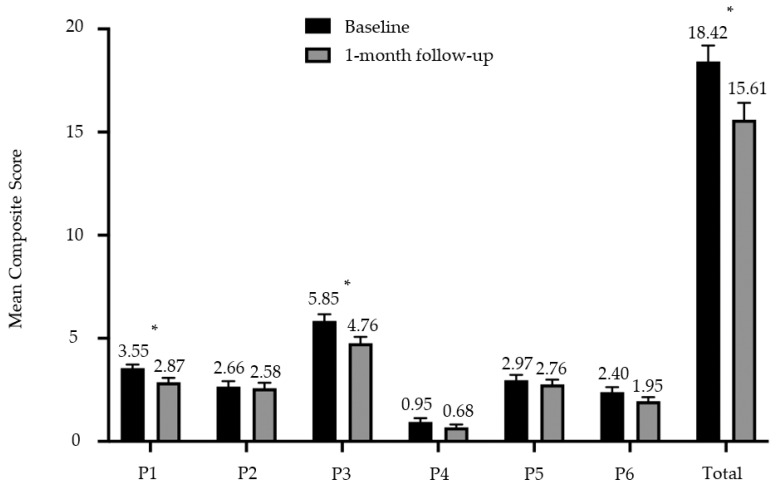
Mean composite score and standard error bar for each P of the 6P intervention tool at baseline and one-month follow-up. * denotes *p* < 0.05.

**Table 1 nutrients-13-04553-t001:** Themes and related quotes from participants for the 6P tool and 6P health messages.

Themes	Subthemes	Examples of Quotes from Participants
**6P tool**		
Perceived acceptability of the 6P tool	Comprehensive content and good language	I think it is quite comprehensive. Because currently I don’t see anything like that yet (05) I like that it’s coloured, (a) coloured copy. So, it is attractive, ‘winning’ all my other mailers. So, it will be enticing for me to fill (it) up…Very straightforward (to understand). I see the example (and) I know (what it means) already (07) Presentation and layout are okay... Language is easy to understand, (I am) able to do it on my own (01)
	Suggestion for clearer definitions of food portion size	…the amount of rice, how do you define (what is) 1/3 of the rice, 2/3 of the rice, one whole plate of rice? …I think it would be better if you go by tablespoon (of rice) (04)
	Mobile application as an ideal delivery platform	Maybe (it can be in) some (form of a) pdf questionnaire then we (can) fill (it and) we just sent it to you all (12) An application (would be good)…because (a) the paper (form) might (be) lost very easily (01) An app that works like a diary, an app is easier to access, if you ask me to record in a book, I don’t think I (will) have the motivation (to fill it up), but (if it is) an app that I can do this (6P tool form) whenever it is convenient (04)
	Weekly or daily frequency of administration is ideal	(The 6P tool) preferably to be (filled) like weekly, so that you can easily recall (your food intake and habit) (13) I don’t mind doing (the 6P tool) daily also so that I can have a feel of what I am eating and when I am supposed to (have my meals) (10)
Perceived relevance and usefulness of the 6P tool	Increased knowledge about healthy eating and more aware of unhealthy eating habits	I mean, the habit is like more to me…especially the questions. I learn the good and bad (aspects of my eating behavior and pattern) … (I learned about) the correct timing (meal timing) (03) Based on this right, the portion… you’re over your 100% already. You have to be aware that you overeat (11). Because (answering) some questions make me feel guilty (about my eating habits), so (because of) that guiltiness, I’ll try to do something to let my guiltiness go off (by changing my behavior)…so definitely, these questions will help me…(to) exercise or eat healthier (15)
	Useful to track their eating habits	They (the 6P tool) can actually sort of like monitor how did we change our intake of food (over time) and things like this (12)
Adjuncts to the 6P tool	Suggestion for a feedback report	I want to see results (diagnosis) and recommendations at the end of filling the tool (07)
	Suggestion for a monitoring chart	I think the logbook is a better way for me to monitor my progress. Its more convenient and keeps me motivated (03) I want some form of summary statistic as feedback (so I know) if I am on the right track or not (04)
**6P mobile health messages**		
Perceived acceptability of mobile health messages	Good content and language	Perfect, all the information is there (02) It explains clearly...it is not very lengthy (12) The language and the concepts (are) easy to follow (13)
	Sending it via a mobile application is ideal	It will be best (delivered) by (a) mobile application (14)
	Weekly frequency of mobile health messages delivery	Weekly will be even better, because information retention would be there (07)
Perceived usefulness of mobile health messages	Serves as a good reminder of their goals	Each of these nudges serves as a reminder (for us to improve our eating habits). Just like my running app, when it is time to run, they will ask me if it is time to go for a run (06)
	Motivates them to improve eating habits	For me, if you have this kind of app or nudge, I feel (more) motivated (to work on my goal) because I know the direction in which to go (and how to improve) (05)
	Provides useful dietary information	I think the most important information that has (been) given (to us through mobile health messages) is the (about the importance of eating) three main meals. I think most of us... feel like I still feel like I should skip meals to lose weight. So, I think this information is good for us to let women know that skipping meals does not mean that it will be a healthier choice (to lose weight) (15)
Suggestions for improving mobile health messages	Include more images	I believe it’s not just words, (adding in) some pictures would be good (12) It will be helpful if there are like a photo that is very localized (focused) (07)
	Messages that list the type of foods to be taken and avoided	(It will be good to include) which food is good to take, which food should be avoided (15)

Notes: The number in brackets indicates the codes of the interviewees.

**Table 2 nutrients-13-04553-t002:** Pearson correlation coefficient (r) for the relationship of overall 6P composite scores with BMI, TFEQ, and IPAQ (*n* = 46).

Measures	*r*	*p*
BMI	0.26	0.086
TFEQ- Cognitive restraint	−0.33	0.027
TFEQ- Disinhibition	0.50	<0.001
TFEQ- Hunger	0.45	0.002
IPAQ- Total MET	^a^ −0.52	0.002

Abbreviation: BMI, Body Mass Index; TFEQ, Three-Factor Eating Questionnaire; IPAQ, International Physical Activity Questionnaire-Short; MET, Metabolic Equivalent Task (MET). ^a^ Spearman’s rank correlation with P5-Physicality composite score.

**Table 3 nutrients-13-04553-t003:** Weight, eating behavior, physical activity, and 6P assessment at baseline and one-month follow-up (*n* = 38).

Variable	Baseline	1-Month Follow-Up	*p* ^a^
Weight (kg)	76.28 ± 11.29	76.37 ± 11.20	0.719
BMI (kg/m^2^)	30.32 ± 4.09	30.36 ± 4.10	0.682
TFEQ-51			
Cognitive restraint	10.26 ± 4.10	11.82 ± 3.83	0.010
Disinhibition	7.63 ± 2.87	6.68 ± 2.90	0.030
Hunger	4.92 ± 2.88	4.50 ± 2.98	0.293
Physical activity based on IPAQ scoring			
Inactive	10 (26.3)	10 (26.3)	0.931
Minimally active	18 (47.4)	17 (44.7)	
High active (HEPA)	10 (26.3)	11 (29.0)	
P1 Portion (amount of carbohydrates per meal, scored from 0–7)	3.00 (2.00–4.50)	2.63 (1.25–3.67)	0.004
0	0	0	0.039
1–2 (recommended)	6 (15.8)	13 (34.2)	
>2	32 (84.2)	25 (65.8)	
P2 Proportion (portion of vegetables per day, scored from 0–100%)	46.88 (6.25–100.00)	40.62 (4.69–100.00)	0.777
<50%	19 (50.0)	20 (52.6)	1.000
≥50% (recommended)	19 (50.0)	18 (47.4)	
P3 Pleasure (total snacks and beverages per day)	2.00 (0.50–3.00)	2.00 (0.50–3.00)	0.601
<3 (recommended)	34 (89.5)	32 (84.2)	0.625
≥3	4 (10.5)	6 (15.8)	
P4 Phase (proportion of daily intake after 7 pm, scored from 0–100%)	25.00 (0–55.00)	20.00 (0–50.00)	0.179
<50% (recommended)	32 (84.2)	34 (89.5)	0.625
≥50%	6 (15.8)	4 (10.5)	
P5 Physicality (total duration per week in mins)	195.00 (60.00–1200.00)	285.00 (75.00–840.00)	0.264
<150	16 (42.1)	13 (34.2)	0.581
≥150 (recommended)	22 (57.9)	25 (65.8)	
P6 Psychology (motivational level, scored from 1–10)	6 (4–8)	6 (3–8)	0.653
≤4	8 (21.1)	4 (10.5)	0.344
≥5 (recommended)	30 (78.9)	34 (89.5)	

Abbreviation: HEPA, Health Enhancing Physical Activity; BMI, Body Mass Index; TFEQ, Three-Factor Eating Questionnaire; IPAQ, International Physical Activity Questionnaire-Short; MET, Metabolic Equivalent Task (MET). ^a^ Based on paired t-test, Wilcoxon sign rank test, or McNemar’s test. Values are presented in *n* (%) for categorical variables and means ± SDs or median (25th–75th percentiles) for continuous variables.

**Table 4 nutrients-13-04553-t004:** 6P diagnosis problem and goal selection at baseline and one-month follow-up.

	Baseline (*n* = 37)		Follow-Up (*n* = 38)	
6P Assessment	Diagnosis	Goal Selection	Diagnosis	Goal Selection
**P1 Portion**	*n* = 36 (97.3%)	*n* = 17 (45.9%)	*n* = 34 (89.5%)	*n* = 18 (47.4%)
Lack of carbohydrate	2 (5.4%)		2 (5.3%)	
Overeating	32 (86.5%)		29 (76.3%)	
Lack of whole grain	17 (45.9%)		15 (40.5%)	
Eating too fast	21 (56.8%)		18 (47.4%)	
**P2 Proportion**	*n* = 35 (94.6%)	*n* = 13 (35.1%)	*n* = 35 (92.1%)	*n* = 15 (39.5%)
Inadequate vegetable & fruit intake	32 (86.5%)		34 (89.5%)	
High fat intake	21 (56.8%)		20 (52.6%)	
**P3 Pleasure**				
**Frequent snacking and unhealthy drink**	*n* = 33 (89.2%)	*n* = 14 (37.8%)	*n* = 31 (81.6%)	*n* = 13 (34.2%)
Frequent snacking	1 (2.7%)		0	
Unhealthy snack and drink	26 (70.3%)		15 (39.5%)	
Mindless snacking	21 (56.8%)		27 (71.1%)	
Alcohol intake	4 (10.8%)		5 (13.2%)	
**Irregular intake**	*n* = 37 (100.0%)	*n* = 8 (21.6%)	*n* = 35 (92.1%)	*n* = 6 (15.8%)
Irregular meal intake	37 (100.0%)		35 (92.1%)	
Meal skipping	28 (75.7%)		22 (57.9%)	
**P4 Phase**	*n* = 21 (56.8%)	*n* = 6 (16.2%)	*n* = 19 (50.0%)	*n* = 6 (15.8%)
Night eating	16 (43.2%)		13 (34.2%)	
Bedtime eating	9 (24.3%)		8 (21.1%)	
**P5 Physicality**	*n* = 11 (29.7%)	*n* = 10 (27.0%)	*n* = 14 (36.8%)	*n* = 10 (26.3%)
Inadequate physical activity	7 (18.9%)		13 (34.2%)	
Activity intensity	4 (10.8%)		1 (2.6%)	
**P6 Psychology**	*n* = 8 (21.6%)	*n* = 3 (8.1%)	*n* = 4 (10.5%)	*n* = 3 (7.9%)
Low motivation	8 (21.6%)		4 (10.5%)	

## Data Availability

The data presented in this study are available on request from the corresponding author.

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
