# Peer review of "Development and Validation of a Lifestyle Behavior Tool in Overweight and Obese Women through Qualitative and Quantitative Approaches"

_nutrients, 2021, doi:10.3390/nu13124553_

Round 1

Reviewer 1 Report

In this manuscript, the authors developed a lifestyle intervention tool, the 6P tool, and evaluated its perceived acceptability and usefulness in overweight and obese women. They showed that there was no significant reduction in weight or increase in physical activity, but significant improvements in cognitive restraint and disinhibition, portion size, pleasure behaviors, and total composite 6P score. Overall, this manuscript is interesting.  

Major concerns:

-The authors used a comprehensive lifestyle intervention system. Please explain the necessity of having such a comprehensive lifestyle intervention tool. For different goals, will different ‘P’ combinations be more efficient than using overall 6P?

-The intervention was carried on for a month, and not much significant change was observed in body weight loss or physical activity. Please indicate the potential improvements can be done for such goals?

-Please provide more information regarding the participants. Do they have history of diabetes or smoking and etc.?

Author Response

Thank you for the constructive comments. We have reviewed them and appended our reply as well as amendments to the manuscript below. We hope you find them satisfactory and review our resubmission favorably.

Response to Reviewers’ Comments

Ref: Manuscript Number: NUTRIENTS-1502983

Thank you for the constructive comments. We have reviewed them and appended our reply as well as amendments to the manuscript below. We hope you find them satisfactory and review our resubmission favorably.

Reviewer 1

  1. The authors used a comprehensive lifestyle intervention system. Please explain the necessity of having such a comprehensive lifestyle intervention tool.

Response: There is no such comprehensive lifestyle intervention tool currently, that is based on a mental model of nutrition and grounded in behavioral science theories. The use of this tool will aim to improve preconception metabolic health, and thus improve fertility, pregnancy outcomes and subsequent health in the long-term. We have added a line in the Introduction to highlight the importance of developing a behavioral intervention tool.

Lines 72-75 (Introduction): In so doing, this comprehensive lifestyle intervention will seek to improve metabolic health and fertility, reduce pregnancy complications, and improve cardiometabolic and neurodevelopmental outcomes in their children.

  1. For different goals, will different ‘P’ combinations be more efficient than using overall 6P?

Response: The different goals were identified based on the diagnoses derived from each “P”. This allows them to work specifically on their goals. The overall 6P composite score, on the other hand, is meant to track their progress over time in a holistic fashion. We have added a line in the Results to emphasize this point.

Lines 340- 342 (Results): A total composite score was then derived by summation of the scores for each P, to provide a simple and convenient summary of the 6P results for both participant and clinician, and track the ongoing progress in their nutritional behaviors.

  1. The intervention was carried on for a month, and not much significant change was observed in body weight loss or physical activity. Please indicate the potential improvements can be done for such goals?

Response: We acknowledge the limitations of a short intervention period, which is unlikely to result in any significant weight loss or increase in physical activity. We have added a line in the discussion to suggest an ongoing sustained approach in improving nutritional behaviors, in order to achieve weight loss and physical activity goals.

Lines 430-432 (Discussion): This ongoing sustained approach of self-monitoring, accompanied by frequent nudges, may be useful in achieving weight loss and other lifestyle goals over time.

  1. Please provide more information regarding the participants. Do they have history of diabetes or smoking and etc.?

Response: Further details on the characteristics of the participants were added in the results section.

Lines 291-293 (Results): Nine participants (19.6%) had an underlying metabolic disorder (diabetes mellitus / hypertension / hyperlipidemia). Most of the participants did not smoke (43/46, 93.5%) or drink alcohol (42/46, 93.3%).

Reviewer 2 Report

The manuscript entitled "Development and Validation of a Lifestyle Behavior Tool in Overweight and Obese Women through Qualitative and Quantitative Approaches" was well-structured, and the topic addressed an interesting and important question from a public health standpoint. Thus, the present manuscript may be of interest to readers of "Nutrients". Moreover, the validation design is sound.

  • As a minor point: I understand that Authors ran properly exact test with no inferential ambitions, nevertheless a causative statistics might have been more appealing once a decent number of participants is enrolled. I wonder whether a brief communication/comment type article would be more appropriate than an actual original article.
  • The introduction the theoretical framework would benefit from citing other studies on similar aspects like:

Vandoni et al.Nutrients2021,13,4459.  https:// doi.org/10.3390/nu13124459

Author Response

Thank you for the constructive comments. We have reviewed them and appended our reply as well as amendments to the manuscript below. We hope you find them satisfactory and review our resubmission favorably.

Reviewer 2

  1. As a minor point: I understand that Authors ran properly exact test with no inferential ambitions, nevertheless a causative statistics might have been more appealing once a decent number of participants is enrolled. I wonder whether a brief communication/comment type article would be more appropriate than an actual original article.

Response: The authors agree that this manuscript should be published as a communication paper.

Line 1: Communication

  1. The introduction the theoretical framework would benefit from citing other studies on similar aspects like: Vandoni et al.Nutrients2021,13,4459.  https:// doi.org/10.3390/nu13124459

Response: As suggested, we have added the two studies to support the introduction and theoretical framework of the Theory of Planned Behavior.

Lines 57-58 (Introduction): Interventions that combine eating and exercise behaviors are often used to alleviate weight problems [3, 9, 10].

  1. Vandoni, M.; Codella, R.; Pippi, R.; Carnevale Pellino, V.; Lovecchio, N.; Marin, L.; Silvestri, D.; Gatti, A.; Magenes, V.C.; Regalbuto, C.; Fabiano, V.; Zuccotti, G.; Calcaterra, V. Combatting Sedentary Behaviors by Delivering Remote Physical Exercise in Children and Adolescents with Obesity in the COVID-19 Era: A Narrative Review. Nutrients 2021, 13, 4459.

Lines 94-96 (Introduction): This is based on the theoretical framework of the Theory of Planned Behavior (TPB) [22] which has been widely used to predict behavioral intention and health-related behaviors [22, 23].

  1. Thompson, N. R.; Asare, M.; Millan, C.;  Umstattd, Meyer M. R.; Theory of Planned Behavior and Perceived Role Model as Predictors of Nutrition and Physical Activity Behaviors Among College Students in Health-Related Disciplines. J Community Health 2020, 45 (5), 965-972.
